# Microarray Analysis Revealed Inflammatory Transcriptomic Changes after LSL60101 Treatment in 5XFAD Mice Model

**DOI:** 10.3390/genes12091315

**Published:** 2021-08-26

**Authors:** Foteini Vasilopoulou, Carmen Escolano, Mercè Pallàs, Christian Griñán-Ferré

**Affiliations:** 1Pharmacology Section, Department of Pharmacology, Toxicology and Therapeutic Chemistry, Faculty of Pharmacy and Food Sciences, Institute of Neuroscience, University of Barcelona (NeuroUB), Av. Joan XXIII 27-31, 08028 Barcelona, Spain; ftn.vasilopoulou@gmail.com (F.V.); pallas@ub.edu (M.P.); 2Laboratory of Medicinal Chemistry (Associated Unit to CSIC), Department of Pharmacology, Toxicology and Medicinal Chemistry, Faculty of Pharmacy and Food Sciences, Institute of Biomedicine (IBUB), University of Barcelona, Av. Joan XXIII, 27-31, 08028 Barcelona, Spain; cescolano@ub.edu

**Keywords:** I2 imidazoline receptors, neuroinflammation, LSL60101, Alzheimer’s disease, transcriptomics, 5XFAD

## Abstract

I2-IR have been found dysregulated in patients with neurodegenerative diseases, such as Alzheimer’s disease (AD), in which the importance of neuroinflammation in the establishment and maintenance of cognitive decline is well-documented. To research the implication of I2-IR in neuroinflammatory pathways altered in AD, we determined the expression profile of genes associated with inflammation in the 5XFAD model treated with LSL60101, a well-established I2-IR ligand. Thus, we performed a qPCR array containing 84 inflammation-related genes. Hierarchical clustering analysis revealed three gene clusters, suggesting that treatment with LSL60101 affects the gene expression associated with inflammation in the 5XFAD model. Furthermore, we evaluated the functions of the three clusters; thereby performing a pathway enrichment analysis using the GO database. As we expected, clusters 2 and 3 showed alterations in the inflammatory response, chemotaxis and the chemokine-mediated signaling pathway, among others. To validate previous results from the gene profiling analysis, the expression levels of a representative subset of mRNAs were selected according to the intensity of the observed changes and their biological relevance. Interestingly, changes induced by LSL60101 in the 5XFAD model were validated for several genes. These results suggest that treatment with LSL60101 in the 5XFAD model reverses the inflammatory process during the development of AD.

## 1. Introduction

Imidazoline receptors (IR) were described in the late 1880s as binding sites for adrenergic ligands such as clonidine, idazoxan and related compounds, but not for adrenaline; thus, they constitute nonadrenergic receptors [1,2,3]. IR have been divided into two classes, I1-IR and I2-IR, primarily based on their sensitivity to clonidine and idazoxan, respectively, and a third atypical imidazoline subtype has also been identified [4,5]. I1-IR activation has been associated with cardiovascular and metabolic effects [6,7,8], whereas I2-IR are widely distributed in the brain and primarily in glial cells [4], and the binding of specific ligands to I2-IR has been shown to induce several pharmacological effects, such as analgesia, anti-inflammatory effects, and neuroprotection [8,9].

Regarding I2-IR primary localization in astrocytes and glial cells, it has been demonstrated that I2-IR ligands modulate glial activity in the mice model brain and in spinal cord injury [10,11]. In these studies, using an experimental autoimmune encephalomyelitis model, 2-BFI administration reduced the expression of inflammatory cytokines, including interferon-g (IFN-γ), tumor necrosis factor-α (TNF-α) and microglial activation. Similarly, in a traumatic injury model, 2-BFI reduced interleukin 1β (IL-1β) secretion and microglia activation [12]. In vitro studies corroborate that I2-IR ligands exert their action on the glial cells by suppressing astrocytic activation induced by lipopolysaccharide (LPS) and decreasing TNF-α levels [13]. Consistent with these changes, the recently described I2-IR ligands MCR5 and B06 decrease the expression of proinflammatory markers, such as TNF-α, IL-1β and interleukin 6 (IL-6), and promote synaptic plasticity in a mouse model of aging and neurodegeneration [14,15,16]. Interestingly, I2-IR have been found dysregulated in patients with neurodegenerative diseases, such as Alzheimer’s disease (AD) [17,18]. AD is a neurodegenerative disease with special histochemical hallmarks, namely amyloid plaques and neurofibrillary tangles [19]. Besides the precise etiology of the pathology, it is well known that neuroinflammation, including the inflammatory levels of cytokines and their corresponding pathways, is the landscape that must be faced in the challenge to identify new therapeutic tools to treat AD [20]. 

The 5XFAD mouse model is a well-characterized double-transgenic APP/PSEN1 model, coexpressing five familial AD mutations. This animal model incorporates AD pathological characteristics, including early plaque formation and gliosis, robust cognitive and behavioral deficits such as memory impairment [21]. The well-established I2-IR ligand LSL60101, first described in 1995 and recently evaluated in depth by our group, presents high selectivity for the I2-IR receptors and an optimal pharmacokinetic and safety profile [22,23]; thus, it has proven to be a useful tool in I2-IR research throughout the years [24,25,26]. Remarkably, LSL60101, similarly to BU224, demonstrated a neuroprotective effect on the familial AD mouse model 5XFAD [27,28]. LSL60101 reduced inflammation inherent to AD and microglial activation in this mouse model, in agreement with the reports mentioned above, further indicating the modulation of inflammatory pathways by I2-IR ligands in the brain and its contribution to glial function and activation.

This work aims to unravel the effect of LSL60101, and consequently the implication of I2-IR receptors in inflammatory pathways related to AD, and, by extension, to other neurodegenerative diseases whose pathologies include neuroinflammation. To this end, we analyzed a set of gene expression panels related to inflammation, including several cytokines and their receptors, regulators, and mediators of signaling pathways and factors implicated in the regulation of immune response after LSL60101 treatment in wildtype and 5XFAD mice. 

## 2. Materials and Methods

### 2.1. Animals and Treatments

The 5XFAD is a double-transgenic APP/PSEN1 that coexpresses AD mutations and presents a robust inflammation background [21]. Seven-month-old female 5XFAD mice (*n* = 23) and wildtype (WT; *n* = 23) mice were used to perform molecular analyses. The animals were randomly allocated to experimental groups and divided into four groups: WT control and 5XFAD control, administrated with vehicle (2-hydroxypropyl)-β-cyclodextrin 1.8%), and WT and 5XFAD treated with I2-IR ligand, administrated with LSL60101 diluted in vehicle (1 mg/kg/day), as shown in Figure 1. Treatment was administered through drinking water for 4 weeks. The animals had free access to food and water and were kept under standard temperature conditions (22 ± 2 °C) and 12 h:12 h light–dark cycles (300 lux/0 lux). The water consumption was controlled each week, and the I2-IR ligand concentration was adjusted accordingly to reach the precise dose.

All studies and procedures for the mouse behavior tests, brain dissection and extractions followed the ARRIVE and standard ethical guidelines (European Communities Council Directive 2010/63/EU) and Guidelines for the Care and Use of Mammals in Neuroscience and Behavioral Research (National Research Council 2003) and were approved by Bioethical Committees from the University of Barcelona (670/14/8102) and the Government of Catalonia (10291, approved 1/28/2018). 

### 2.2. Brain Processing and RNA Extraction

Mice were euthanized by cervical dislocation after the treatment period. The brains were immediately removed from the skulls, and the hippocampi were dissected, frozen and maintained at −80°C. Total RNA isolation from hippocampal samples was performed using the TRIzol® reagent according to the manufacturer’s instructions (Bioline Reagent, London, UK). The yield, purity and quality of RNA were determined spectrophotometrically with a NanoDrop™ND-1000 apparatus (Thermo Fisher, Waltham, MA, USA) and an Agilent 2100B Bioanalyzer (Agilent Technologies, Santa Clara, CA, USA). RNA samples with 260/280 ratios and RINs higher than 1.9 and 7.5, respectively, were selected. A reverse transcription-polymerase chain reaction (RT-PCR) was performed. Briefly, 1 μg and 2 μg of messenger RNA (mRNA) were reverse transcribed using a high-capacity cDNA reverse transcription kit (Applied Biosystems, Foster City, CA, USA) for PCR array performance and q-PCR validation, respectively.

### 2.3. Real-time Quantitative PCR Array

A real-time quantitative PCR array containing 84 inflammation-related genes (qPCR Sign Arrays 96 system, AnyGenes, Paris, France) was used for screening according to the instructions of the manufacturer. Briefly, 2 μL of diluted cDNA pooled samples (*n* = 4; 2 μg cDNA diluted at 1/12 from reverse transcription (20 μL) performed with 1 μg of RNA) was mixed with 10 μL of 2× Perfect Master Mix SYBR Green and 8 μL ultrapure H_2_O and added to each well; consequently, the total reaction volume was 20 μL per well. After 20 μL of the reaction mix was in each well of the 96-well plate, the plate was centrifuged, and then the qPCR run was performed using a Step One Plus Detection System (Applied-Biosystems), following the manufacturer’s recommendations and protocols. PCR reaction conditions were 95 °C, 10 min; 95 °C, 5 s and 60 °C, 30 s, ×40 cycles. After completion of the reaction, the melting curve was analyzed, 95 °C, 10 s, 65–95 °C, 30 s.

### 2.4. Hierarchical Clustering

We performed a hierarchical clustering with the genes analyzed in the qPCR Sign Arrays 96 system to evaluate the expression profile between the study groups. These genes were clustered into three groups based on the expression profile using the R package heatmap. The expression data were clustered by Euclidean distances between genes and by applying the complete method for hierarchical clustering. Complete microarray gene expression data are presented in Appendix A.

### 2.5. Protein–Protein Interaction Network and Functional Annotation

To determine the interactions between the groups, we performed protein–protein interaction networks using the database STRING [29]. A PPI enrichment *p*-value < 0.001 was considered statistically significant, indicating that the proteins are at least partially biologically connected. To determine the functional annotation of the three groups, we determined the gene ontology (GO) and performed pathway analysis with the Kyoto Encyclopedia of Genes and Genomes (KEGG), using the Database for Annotation, Visualization, and Integrated Discovery (DAVID) [30]. GO terms and KEGG pathways with an adjusted *p*-value < 0.05 were considered statistically significant. We used the KEGG mapping tool to display the downregulated (green) and upregulated genes (red) in KEGG pathway maps [31]. To evaluate the transcriptional regulatory interactions between the three groups of genes and mouse transcriptional factors (TFs), we used the TRRUST database [32]. TRRUST identifies potential TFs involved in the regulation of genes of interest. TFs with an adjusted *p*-value < 0.05 were considered statistically significant.

### 2.6. Gene Expression Validation with Real-Time Quantitative PCR

To confirm the PCR array results, which identified specific genes as responding to I2-IR treatment, quantitative SYBR® Green real-time PCR was performed using a Step One Plus Detection System (Applied-Biosystems) with SYBR® Green PCR Master Mix (Applied-Biosystems). Each reaction mixture contained 6.75 μL of complementary DNA (cDNA) (with a concentration of 2 μg), 0.75 μL of each primer (with a concentration of 100 nM) and 6.75 μL of SYBR® Green PCR Master Mix (2×). 

The data were analyzed utilizing the comparative cycle threshold (Ct; ΔΔCt) method, in which the levels of a housekeeping gene are used to normalize differences in sample loading and preparation. Normalization of expression levels was performed with β-actin. The primer sequences used are presented in Appendix A. Each sample was analyzed in duplicate, and the results represent the ratio percentage of the transcript levels among different groups compared to the control group.

### 2.7. Statistical Analysis

Data analysis was conducted using GraphPad Prism ver. 8 statistical software. Data are expressed as the mean ± standard error of the mean (SEM) of 5–6 samples per group. All data were tested for normal distribution and equal variance. Means were compared with two-way analysis of variance (ANOVA) followed by the Tukey post hoc test. Comparison between groups was also performed by a two-tailed Student’s *t*-test for independent samples when it was necessary. Statistical significance was considered when *p* values were < 0.05. The statistical outliers were determined with Grubs’ test and when necessary were removed from the analysis. 

## 3. Results

### 3.1. LSL60101 Treatment Regulates Genes Associated with the Inflammatory Response

To determine the expression profile of genes associated with inflammation in the 5XFAD model treated with LSL60101, we performed a real-time quantitative PCR array containing 84 inflammation-related genes. Hierarchical clustering analysis revealed three gene clusters (Figure 2A). Interestingly, cluster 2 was characterized by genes with reduced expression after LSL60101 treatment (Figure 2A). Cluster 3 had increased expression of genes after treatment in the 5XFAD mice (Figure 2A). These results suggest that treatment with LSL60101 affects the gene expression associated with inflammation in the 5XFAD model.

To evaluate the functions of the three clusters, we performed a pathway enrichment analysis using the GO database. As we expected, clusters 2 and 3 show alterations in processes, such as the inflammatory response, chemotaxis and the chemokine-mediated signaling pathway (Figure 2B), indicating that these mechanisms are involved in the effects observed after treatment with LSL60101 in 5XFAD mice. Additionally, KEGG analysis demonstrated alteration in the chemokine signaling pathway and cytokine–cytokine receptor interaction (Figure 3A,B). Notably, in the pathway of Figure 3B, among the deregulated genes detected in response to treatment with LSL60101, we can observe a reduction in *Tnf-α* and *Il-6*, two cytokines with high expression in the development of AD [33]. These results suggest that treatment with LSL60101 reverses some of the inflammatory genes related to cognitive decline in the 5XFAD model.

### 3.2. NF-κβ Pathway Regulates the Inflammatory Response in LSL60101 Treatment

NF-κβ is a transcription factor that regulates multiple aspects associated with inflammatory responses [34]. Using the TRRUST database, a manually curated database of human and mouse transcriptional regulatory networks, we found that the genes present in clusters 2 and 3 can be regulated by RELA and NFKB1 (Figure 4A), two subunits of the transcription factor NF-κβ [35], suggesting that NF-κβ regulates the neuroinflammation process in the 5XFAD model after treatment. Interestingly, several members of the NF-κβ signaling pathway, such as IL-1β, cyclooxygenase 2 (COX2), TNF-α and MIP-2, were reduced in the treatment group (Figure 4B). Altogether, we suggest that treatment with LSL60101 alters the expression of genes associated with neuroinflammation processes in the 5XFAD model, which could be dependent on the NF-κβ pathway, reinforcing the role of this pathway as a therapeutic target for AD [35].

### 3.3. Validation of a Representative Subset of Genes Involved in Neuroinflammation and AD

To validate previous results from the gene profiling analysis, the expression levels of a representative subset of mRNAs were selected according to the intensity of the observed changes and their biological relevance and were measured by single real-time PCR in the hippocampus samples from each group. We evaluated the expression of C*-X-C motif chemokine receptor 2 (Cxcr2), Toll-like receptor 5 (Tlr5), CD40 ligand (CD40lg), chemokine (C-C motif) ligand 7 (Ccl7), C-C chemokine receptor type 4 (Ccr4), Ifn-**γ, E-selectin (Sele), chemokine (C-C motif) ligand 12 (Ccl12), chemokine (C-C motif) ligand 8 (Ccl8)* and *C-X-C motif chemokine ligand 10 (Cxcl10)* (Figure 5A–J). Interestingly, changes induced by LSL60101 in the 5XFAD model were validated for *Cxcr2, Tlr5* and *Sele* (Figure 5A,B,G), and a clear decreasing tendency in the 5XFAD LSL60101-treated group for *CD40lg, Ccl7* and *Ccr4* was observed (Figure 5C–E). On the contrary, an increasing tendency in the 5XFAD LSL60101-treated group was found for *Ccl12*, *Ccl8* and *Cxcl10* (Figure 5F,H–J). Similarly, changes induced by LSL60101 in the WT model were not statistically significant; however, a decreasing tendency for *CD40lg, Ccl8* and *Cxcl10* (Figure 5C,I,J) and an increasing tendency for *Ccr4* and *Ccl7* were observed (Figure 5D–E). Significant changes in the WT LSL60101- and 5XFAD LSL60101-treated groups were found for *CD40lg, Ccl12, Ccl8* and *Cxcl10* (Figure 5C,H–J).

## 4. Discussion

As aforementioned, AD is characterized by multiple molecular signatures at different stages of the disease, neuroinflammation being one of the most relevant early events in the disease [36]. Indeed, chronic inflammation contributes to neuronal dysfunction and cognitive decline [37]. In the current AD drug development pipeline, inflammation is addressed by several drugs with different action mechanisms, demonstrating its potential as a major target for effective AD treatment [38]. Evidence provided by our group and others demonstrates that selective I2-IR ligands can modulate neuroinflammation, promoting changes in inflammatory cytokine protein levels and/or gene expression in AD mice models [14,16,27,39]. Nevertheless, researching the specific mechanisms whereby I2-IR ligands modulate inflammatory pathways is necessary to understand better the potential link between their neuroprotective and anti-inflammatory effects. Thus, the main goal of this work was to study and identify new inflammatory transcriptome biomarkers after LSL60101 treatment and validate some of the inflammatory markers that were found altered in our previous works. Here, we used the hippocampal transcriptome of 5XFAD mice, which harbor five APP/PSEN1 mutations, leading to a robust Aβ production and deposition in their brains [21]. Interestingly, we recently demonstrated that chronic treatment with LSL60101 improved cognitive impairment and reduced Aβ plaques and tau pathology in 7-month-old 5XFAD mice [28].

In the present study, a gene expression profile study of the hippocampus samples from the 5XFAD mice indicated that LSL60101 treatment significantly modified the expression of several genes, generating three hierarchical clusters based on the enrichment heatmap. Not surprisingly, two clusters were characterized by genes with reduced expression after LSL60101 treatment, suggesting an anti-inflammatory effect of LSL60101. In line with these results, in the AD landscape provided by the 5XFAD mice, LSL60101 was shown to decrease microglial and astroglial reactivity by reducing Iba-1 and GFAP levels, respectively [28]. Indeed, microglia and astrocytes are central players in the neuroinflammatory process, producing proinflammatory or anti-inflammatory cytokines upon pathological insults [40]. Similarly, one hierarchical cluster was characterized by increased anti-inflammatory gene expression in the LSL60101 group, confirming our hypothesis of anti-inflammatory properties of the selective I2-IR ligand. Notably, 17 genes in the hippocampus of LSL60101-treated mice displayed a significant change of more than twofold in their expression, and we were able to validate the modifications for *Ccr4, Ifn-γ* and *Sele* (cluster 2) and *Cxcr2, Tlr5, CD40lg, Ccl7, Ccl12, Ccl8* and *Cxcl10* (Cluster 3). Among them, *Cxcr2, Tlr5, Sele, CD40lg, Ccl7* and *Ccr4* were found to significantly decrease or presented a clear tendency to decrease after treatment, while *Ccl8, Ccl12* and *Cxcl10* gene expression tended to increase after treatment in 5XFAD mice. 

Thus, our results indicated an anti-inflammatory landscape, with several reduced proinflammatory and several increased anti-inflammatory cytokines. For instance, the receptor CXCR2 presents a prominent expression at microglia in AD compared to the normal brain tissue and could be used as a strategic therapeutic target to counterbalance inflammatory microenvironments in AD [41,42]. On the other hand, it has been suggested that the CD40–CD40LG interaction may be involved in the inflammatory pathways in AD. It has been demonstrated that CD40LG and Aβ synergistically increase TNF-α and promote neuronal death, reinforcing the AD pathology [43]. Finally, overexpression of different chemokine receptors, including CCR4, has been identified in T-cells of AD patients, linking these inflammatory cells to brain damage [44]. Taken together, the results suggest that treatment with IR-I2 ligands as LSL60101 might attenuate the neuroinflammatory process in AD by reversing the expression of inflammatory mediators.

Moreover, as we expected, our results showed alterations in GO enrichment analysis in processes such as the inflammatory response, chemotaxis and the chemokine-mediated signaling pathway. Similarly, KEGG pathways are related to chemokine signaling pathways, cytokine–cytokine receptor interaction and Toll-like receptor signaling, among others. Altogether, this evidence suggests that the selective I2-IR ligand, LSL60101, presents diversity in the modulation of pathways and biological functions in AD. In accordance with our results, the selective I2-IR ligand has been shown to mediate pleiotropic central effects in vivo and in vitro, including alterations in dopamine and serotonin levels, acute hyperphagic effects, inhibition of the development of the opioid-induced tolerance, potentiation of morphine analgesia, glia modulation and neuroprotection [22,24,45,46,47,48]. Interestingly, gene set enrichment analysis identified several genes such as *Il-1β*, *Cox2*, *Tnf-α* and *Mip-2*, which were present in clusters 2 and 3 associated with the NF-κβ pathway and were reduced in the LSL60101 treatment group. Of note, activation of the NF-κβ pathway is closely related to neurodegeneration and particularly to AD [35], while its inhibition has been shown to improve cognitive deficits in in vivo models of AD [49]. In turn, reductions in NF-κβ-regulated genes, such as *Il-1β*, *Cox2* and *Tnf-α*, were observed after treatments with selective I2-IR ligands, delivering neuroprotection in mouse models of aging neurodegeneration and AD [14,15,16,39]. Therefore, downregulation of the NF-κβ pathway by LSL60101 might partially account for the altered gene expression of inflammatory markers and further explain its neuroprotective effect in this AD mouse model. In this line of evidence, NF-κβ has been implicated in APP processing and facilitation of Aβ generation [50]; thus, its downregulation is in accordance with the amelioration of Aβ pathology induced by LSL60101 treatment in 5XFAD mice.

In conclusion, our study identified several genes, modulated pathways, transcriptional factor signatures and their transcriptional regulatory networks that correlated with previously reported cognitive improvement after LSL60101 treatment in 5XFAD mice [28], demonstrating the potential of selective I2-IR ligands for AD treatment.

## Figures and Tables

**Figure 1 genes-12-01315-f001:**
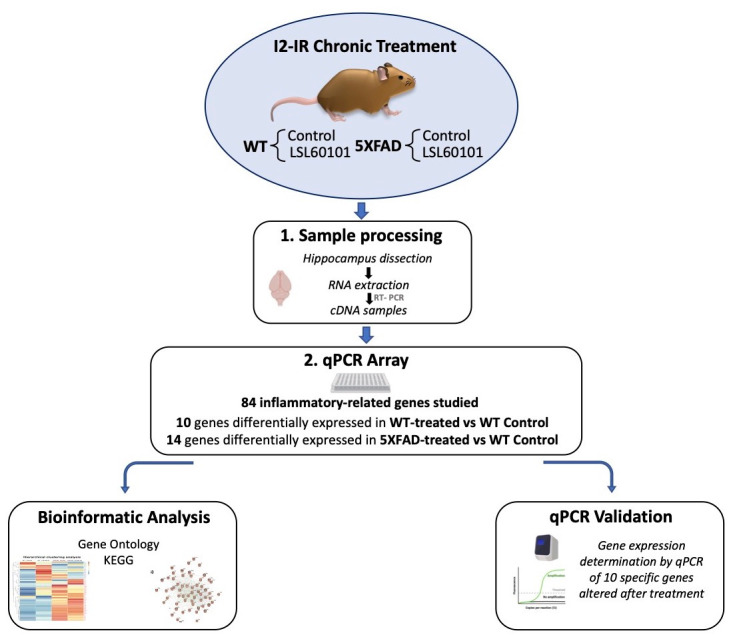
Experimental design of sampling processing, bioinformatic analysis and qPCR validation.

**Figure 2 genes-12-01315-f002:**
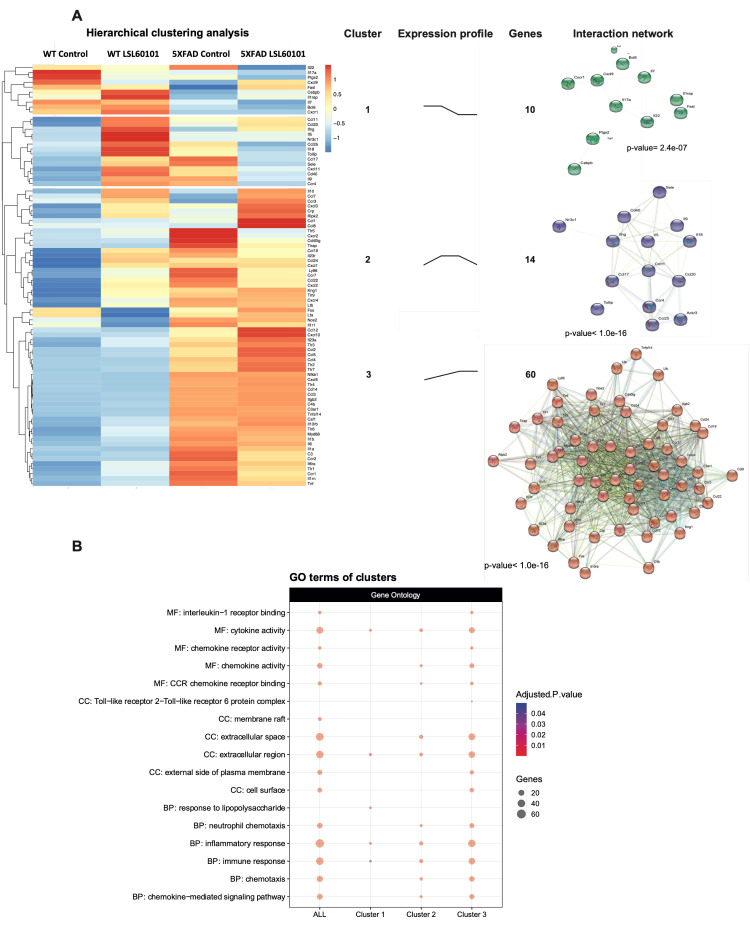
Hierarchical clustering and gene ontology of inflammatory genes in WT and 5XFAD mice treated with LSL60101. (**A**) The genes present in the qPCR array were subjected to hierarchical clustering analysis. The heatmap shows three clusters with a specific expression profile between the experimental conditions represented in "expression profile". The predicted protein–protein interactions analysis is shown, as well as the PPI enrichment *p*-value of each cluster. (**B**) The dotplot shows the top GO terms of the clusters. An adjusted *p*-value < 0.05 was considered statistically significant.

**Figure 3 genes-12-01315-f003:**
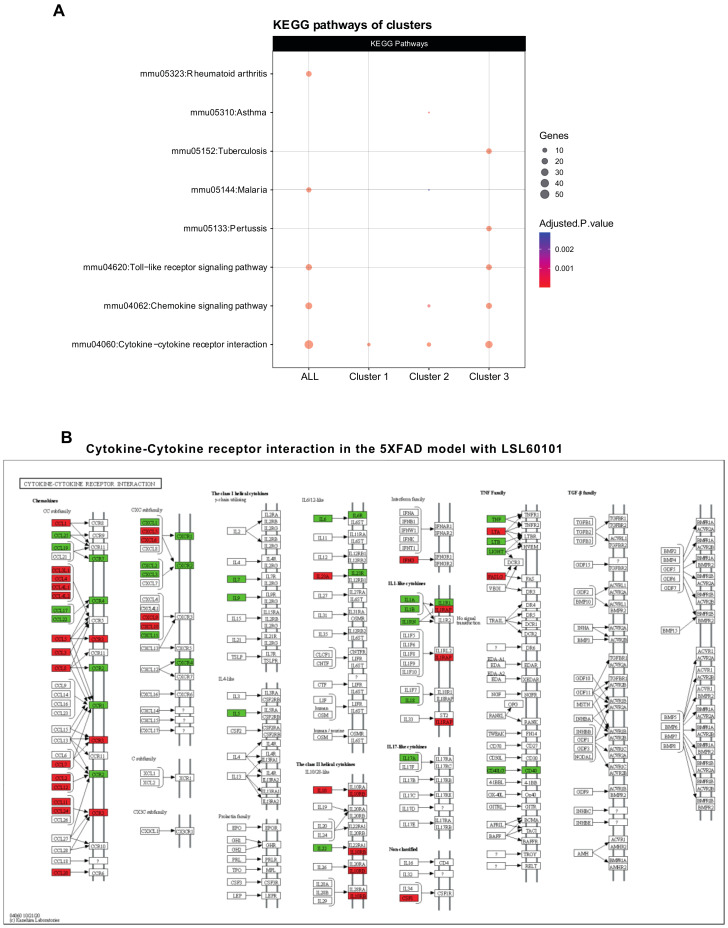
KEGG pathway analysis and cytokine–cytokine receptor interaction pathway. (**A**) The dotplot shows the top KEGG pathways of the clusters. The cytokine–cytokine receptor interaction is representative of the three clusters. An adjusted *p*-value < 0.05 was considered statistically significant. (**B**) The KEGG pathway map cytokine–cytokine receptor interaction (mmu04060) is shown. The downregulated (green) and upregulated genes (red) in the 5XFAD control condition versus the 5XFAD LSL60101 condition are shown.

**Figure 4 genes-12-01315-f004:**
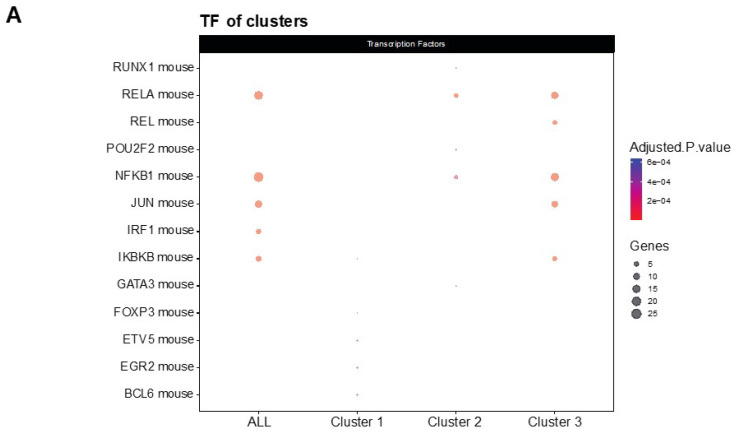
Transcriptional regulatory interactions analysis and NF-κβ signaling pathway. (**A**) The dotplot shows the top TFs involved in the regulation of the clusters identified with TRRUST. An adjusted *p*-value < 0.05 was considered statistically significant. (**B**) The KEGG pathway map NF-κβ signaling pathway (mmu04064) and the adjusted *p*-value of the significant clusters are shown. The downregulated (green) and upregulated genes (red) in the 5XFAD control condition versus the 5XFAD LSL60101 condition are shown.

**Figure 5 genes-12-01315-f005:**
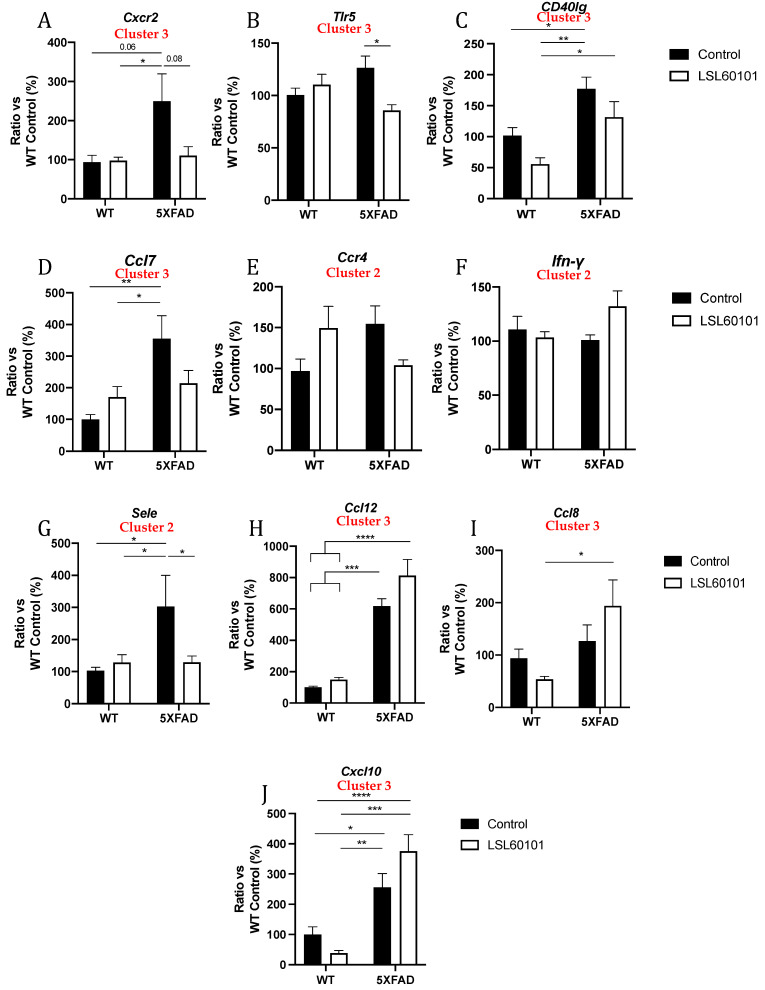
Validation of representative subset of mRNAs. Gene expression of *Cxcr2, Tlr5, Cd40lg, Ccl7, Ccr4, Ifn-γ, Sele, Ccl12, Ccl8* and *Cxcl10* (**A**–**J**) in the hippocampus of WT and 5XFAD control and LSL60101-treated mice. Gene expression levels were determined by real-time PCR. Values in bar graphs are adjusted to 100% for relative gene expression of the WT control. Bars represent mean ± SEM. Two-way ANOVA with Tukey post hoc analysis, * *p* < 0.05; ** *p* < 0.01; *** *p* < 0.001; **** *p* < 0.0001; *n* = 5–6 per group.

## Data Availability

Not applicable.

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
