# Peer review of "Microarray Analysis Revealed Inflammatory Transcriptomic Changes after LSL60101 Treatment in 5XFAD Mice Model"

_genes, 2021, doi:10.3390/genes12091315_

Round 1

Reviewer 1 Report

In this manuscript, Vasilopoulou and colleagues assess the expression of inflammatory-related genes in the control wild-type and the 5XFAD Alzheimer’s disease mouse model after I2-1R agonist LSL60101 treatment. It has been shown that LSL60101 treatment reduces several AD-associated pathologies in 5XFAD mouse model (Vasilopoulou et al., Br J Pharmacol., 2021), yet the levels of inflammatory-associated genes have not been characterized in this model. By using qPCR array, the authors show that LSL60101 treatment is able to reduce a subset of inflammatory genes, and many of which are involved in cytokine-cytokine receptor pathway and regulated by NF-kB signaling. Overall, the manuscript is topically relevant.

Specific comments:

(1) All the figures shown in the current manuscript are poorly presented, and I suggest the authors should really make efforts on improving it.

a> The figure in page 3 is missing, and it should be Figure 1 not Figure 2.

b> It would be helpful to increase the font size for Figure 2A and 3B.

(2) Line 197-199, “These results suggest that treatment with LSL60101 in the 5XFAD model reverses the inflammatory process during the development of AD.” The authors should modify their statement as LSL60101 treatment only reverses the expression of some but not all the inflammatory-related genes in qPCR array experiment.

(3) In the qPCR validation experiment, it is noted that Cxcl10 level in Control group is not 100%. Please correct it.

(4) Line 114-115 “RNA samples with 260/280 ratios and RINs higher than 7.5, respectively”, the value for 260/280 ratios is missing.

(5) References cited in the section of Protein-protein interaction network and functional annotation are not listed in the reference section.

(6) In the Statistical Analysis section, the authors stat that “Data are expressed as the mean ± Standard error of the mean (SEM) of at least 6 samples per group.” It is not correct as some experiments only included less than 6 data points.

Author Response

We thank all comments and suggestions of the reviewers about the manuscript submitted to the Genes Journal.

We did our best to complete the information required by answering point by point their concerns.

Reviewer 1

In this manuscript, Vasilopoulou and colleagues assess the expression of inflammatory-related genes in the control wild-type and the 5XFAD Alzheimer’s disease mouse model after I2-1R agonist LSL60101 treatment. It has been shown that LSL60101 treatment reduces several AD-associated pathologies in 5XFAD mouse model (Vasilopoulou et al., Br J Pharmacol., 2021), yet the levels of inflammatory-associated genes have not been characterized in this model. By using qPCR array, the authors show that LSL60101 treatment is able to reduce a subset of inflammatory genes, and many of which are involved in cytokine-cytokine receptor pathway and regulated by NF-kB signaling. Overall, the manuscript is topically relevant. 

Specific comments:

(1) All the figures shown in the current manuscript are poorly presented, and I suggest the authors should really make efforts on improving it.

a> The figure in page 3 is missing, and it should be Figure 1 not Figure 2.

b> It would be helpful to increase the font size for Figure 2A and 3B.

We thank the reviewer for noticing these. We have corrected the errors labelling and increase the font size in the Figure 2A and 3B to improve readability. Likewise, we have revised the missing figures, we believe it was a problem generated by the word file due to the figure are too bigger.

(2) Line 197-199, “These results suggest that treatment with LSL60101 in the 5XFAD model reverses the inflammatory process during the development of AD.” The authors should modify their statement as LSL60101 treatment only reverses the expression of some but not all the inflammatory-related genes in qPCR array experiment.

We thank the reviewer for this comment and we modified the statement in the main text accordingly. In particularly we substituted for “ These results suggest that treatment with LSL60101 reverses some of the inflammatory genes related to cognitive decline in the 5XFAD model”.

(3) In the qPCR validation experiment, it is noted that Cxcl10 level in Control group is not 100%. Please correct it.

 Thank you for noticing this. We apologize for this error. We corrected the bar chart for Cxcl10 in fig 5. 

(4) Line 114-115 “RNA samples with 260/280 ratios and RINs higher than 7.5, respectively”, the value for 260/280 ratios is missing.

 We added the missing value (1.9) in the section. 

(5) References cited in the section of Protein-protein interaction network and functional annotation are not listed in the reference section.

We thank the reviewer for this observation. Accordingly, we added the missing references in the reference list.  

  1. Damian Szklarczyk, Annika L Gable, Katerina C Nastou, David Lyon, Rebecca Kirsch, Sampo Pyysalo, Nadezhda T Doncheva, Marc Legeay, Tao Fang, Peer Bork, Lars J Jensen, Christian von Mering, The STRING database in 2021: customizable protein–protein networks, and functional characterization of user-uploaded gene/measurement sets, Nucleic Acids Research, Volume 49, Issue D1, 8 January 2021, Pages D605–D612,
  2. Dennis, G., Sherman, B.T., Hosack, D.A.et al. DAVID: Database for Annotation, Visualization, and Integrated Discovery. Genome Biol 4, R60 (2003). https://doi.org/10.1186/gb-2003-4-9-r60.
  3. Dai Z, Tang H, Pan Y, Chen J, Li Y, Zhu J. Gene expression profiles and pathway enrichment analysis of human osteosarcoma cells exposed to sorafenib. FEBS Open Bio. 2018;8(5):860-867. Published 2018 Apr 24. doi:10.1002/2211-5463.12428.
  4. Han H, Cho JW, Lee S, et al. TRRUST v2: an expanded reference database of human and mouse transcriptional regulatory interactions. Nucleic Acids Res. 2018;46(D1):D380-D386. doi:10.1093/nar/gkx1013.

(6) In the Statistical Analysis section, the authors stat that “Data are expressed as the mean ± Standard error of the mean (SEM) of at least 6 samples per group.” It is not correct as some experiments only included less than 6 data points.

We thank the reviewer for this comment. In the experiments were used 5-6 samples per group, so we changed this accordingly in the main text.

Reviewer 2 Report

It is an interesting paper. I have a few comments, which may improve the manuscript.

  1. In introduction, could you explain on I1-IR too? A brief explanation would be enough.
  2. Also in introduction, could you introduce LSL60101 a little more?A short introduction would be useful. 
  3. In methods part, Figure 2 (or Figure 1?.) on experimental design can not be displayed. Could you upload this figure again? 
  4. Also in methods, (Real time quantitative PCR paragraph), could you add a supplementary file on the list of 84 inflammatory genes, which were analyzed in this study?

Author Response

We thank all comments and suggestions of the reviewers about the manuscript submitted to the Genes Journal.

We did our best to complete the information required by answering point by point their concerns.

Reviewer 2

It is an interesting paper. I have a few comments, which may improve the manuscript.

  1. In introduction, could you explain on I1-IR too? A brief explanation would be enough.

A short mentioning in I1-IR was added in the introduction. Lines 39-40

  1. Also in introduction, could you introduce LSL60101 a little more?A short introduction would be useful. 

A short introduction about the compound LSL60101 was added in lines 66-70.

  1. In methods part, Figure 2 (or Figure 1?.) on experimental design can not be displayed. Could you upload this figure again? 

We apologize for this mistake. We corrected the figure enumeration accordingly.

  1. Also in methods, (Real time quantitative PCR paragraph), could you add a supplementary file on the list of 84 inflammatory genes, which were analyzed in this study?

We thank reviewer for noticing these. Accordingly, we have added in each section methods the sentence for the primer list: “The primer sequences used are presented in Supplementary Table S1” and the complete array data: “Furthermore, complete microarray gene expression data are presented in Supplementary Table S2”, with their corresponding supplementary file.
